# Towards higher frequencies in a compact prebunched waveguide THz-FEL

Andrew Fisher [1], Maximilian Lenz[1], Alex Ody[1], Yining Yang [1], Chad Pennington[2], Jared Maxson[2], Tara Hodgetts[3], Ronald Agustsson [3], Alex Murokh[3] & Pietro Musumeci [1] ✉

Free-electron-lasers fill a critical gap in the space of THz-sources as they can reach high average and peak powers with spectral tunability. Using a waveguide in a THz FEL significantly increases the coupling between the relativistic electrons and electromagnetic field enabling large amounts of radiation to be generated in a single passage of electrons through the undulator. In addition to transversely confining the radiation, the dispersive properties of the waveguide critically affect the velocity and slippage of the radiation pulse which determine the central frequency and bandwidth of the generated radiation. In this paper, we characterize the spectral properties of a compact waveguide THz FEL including simultaneous lasing at two different frequencies and demonstrating tuning of the radiation wavelength in the high frequency branch by varying the beam energy and ensuring that the electrons injected into the undulator are prebunched on the scale of the resonant radiation wavelength.

In recent years, there has been a growing interest in the development of terahertz sources (0.1–10 THz) for both scientific research and industrial applications, including time-domain spectroscopy for solid-state systems and low density plasmas, high-gradient high-frequency acceleration, terahertz-based imaging for medical and security applications, and electron paramagnetic resonance[1–6]. In addition to common compact sources such as quantum cascade lasers (QCLs), gyrotrons, and laser-based sources[7–9], free electron lasers (FELs) have begun to emerge as an attractive source due to their frequency tunability, temporal coherence, and high peak power with repetition rates constrained only by the availability of electron sources. Numerous accelerator THz-FEL facilities exist[10–13] and there is particular interest in developing facilities with THz-Xray pump probe capabilities[14,15].

The single pass gain of an FEL operating at long wavelengths is attenuated by radiation diffraction and slippage effects. A waveguide can be used to transversely confine the field along the undulator, and the slippage can be eliminated by choosing waveguide dimensions such that the subluminal radiation group velocity is matched to the average longitudinal electron beam velocity[16]. As already recognized in the early '90s[17], the introduction of the waveguide and the zero slippage condition enables the use of RF-compressed, high current density beams, short compared to the radiation wavelength, for seeding and the generation of large-bandwidth, few cycle THz pulses with intense fields. In this regime, the gain curve is significantly broader than a typical free-space FEL, the interaction length is extended, and 10% energy extraction efficiency has been previously demonstrated[18].

The additional constraint on experimental parameters restricts zero-slippage resonance to one single frequency for a given undulator and waveguide aperture, undermining FEL tunability. It also places tight tolerances on transverse alignment and matching of the beam through the undulator as the wiggling trajectory amplitude becomes comparable with the waveguide size, making it challenging to transmit the full charge through the system. To further extend the frequency range accessible by the FEL, it is possible to purposefully detune the beam energy from the zero-slippage condition, striking a compromise between maximal tunability and minimal slippage in the interaction. Increasing the beam energy has the additional advantages of easing the charge transmission in the system as well as increasing the frequency of the generated radiation.

[1]Department of Physics and Astronomy, UCLA, 405 Hilgard Avenue, Los Angeles, CA, USA. [2]Department of Physics, Cornell University, 616 Thurston Ave., Ithaca, NY, USA. [3]RadiaBeam Technologies, 1717 Stewart St., Santa Monica, CA, USA. ✉e-mail: musumeci@physics.ucla.edu

In this regime off the zero-slippage condition, though, the strategy of fully compressing a beam to sub-wavelength scale is limited by strong space charge forces as well as the slippage introduced by the group velocity mismatch. In order to preserve the compactness of the system, effective seeding for the FEL interaction can be obtained by injecting a multicycle-long beam with strong spectral form factor at the system's resonant wavelength[19]. Over the years, various schemes have been developed to control the longitudinal beam profile and generate microbunching at the ps-scale, including photocathode laser shaping[20,21], energy modulation of the beam on the ps-time scale[22,23] and dispersive section masking[24,25]. For high current applications ranging from coherent radiation generation to wakefield acceleration, it has been shown that space charge effects can be used to effectively enhance the bunching content by controlling the evolution of the space-charge induced beam plasma oscillations[26,27].

In this paper, we investigate the operation of a waveguide THz-FEL at energies above the zero-slippage condition at the UCLA Pegasus photoinjector. A permanent magnet chicane is used to compress the beam and seed the interaction in a 1-meter long helical undulator. Direct measurement of the electromagnetic field by electro-optic sampling reveals broadband lasing in two frequency branches which propagate with different group velocities in the system. The low frequencies (~120 GHz) are bound below by the nominal TE11 4.05 mm diameter waveguide cutoff frequency while the high frequency branch can be tuned in the range 500–700 GHz by varying the beam energy between 7.5 and 8.5 MeV. We also show that by taking advantage of photoinjector laser shaping with birefringent $\alpha$-BBO crystals, it is possible to seed with a beamlet distribution and enhance the high frequency content in the radiation. The manuscript is organized as follows. First, we begin by detailing the physics of the zero-slippage interaction detuned from group resonance. Next, we introduce the experimental beamline specifically focusing on beam generation and the compact permanent magnet chicane. Finally, we discuss the THz diagnostics and present experimental data demonstrating improved high frequency generation and successful frequency tuning achieved through beam prebunching.

## Results

### Zero-slippage resonance

The FEL interaction produces radiation at a frequency satisfying the phase resonance condition

$$k_z + k_u = \frac{\omega}{c\beta_z} \tag{1}$$

where $k_z$ is the longitudinal wavevector, $k_u = 2\pi/\lambda_u$ where $\lambda_u$ is the undulator period, $\omega$ is the resonant angular frequency, and $c\beta_z$ is the average longitudinal beam velocity. This requires that radiation wavefronts slip ahead of the electrons by one wavelength each undulator period. The waveguide dispersion, $\omega^2/c^2 = k_z^2 + k_\perp^2$, alters the phase and group velocity of the radiation according to $v_p = \omega/k_z$ and $v_g = c^2 k_z/\omega$ where $k_\perp = 1.8412/R$ for the TE11 circular waveguide mode with radius $R$. The zero-slippage condition ($v_g = c\beta_z$) is achieved by choosing waveguide dimensions to match the subluminal group velocity of the radiation to the average longitudinal velocity of the electron beam. The wavefronts, traveling at superluminal phase-velocity, continue to wash over the electrons according to Eq. (1), but the temporal envelope remains aligned with the beam allowing ultrashort bunches to continuously interact with the radiation generating few cycle pulses with intense fields. With minimal slippage, we can take advantage of short, high current beams to greatly increase the FEL gain allowing for high-efficiency compact sources. Both undulator field strength and waveguide parameters could in principle be tapered to maintain resonance with the decelerating electrons[28] along the interaction. For clarity, we refer to the zero-slippage

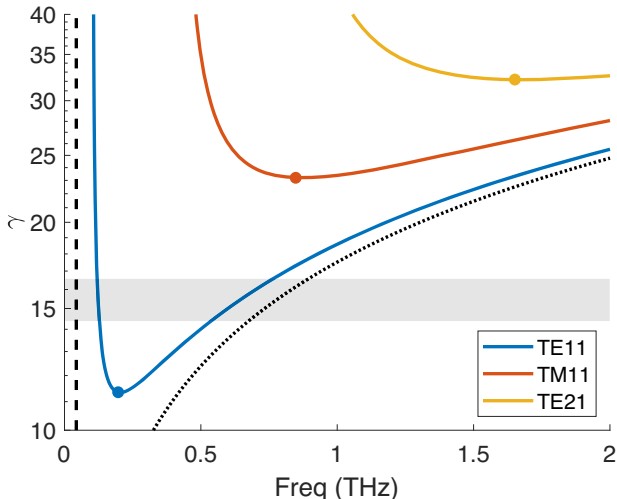

**Fig. 1 | Phase resonance curves of the 3 lowest modes for the 4.05 mm diameter waveguide.** Markers indicate group resonance and dashed/dotted lines show the bounding fundamental cutoff frequency and free-space limit, respectively. The beam energy range of the experiment is shaded.

condition and Eq. (1) as group resonance and phase resonance, respectively.

For a given waveguide, the zero-slippage condition is satisfied at a singular frequency thereby limiting wide-range FEL tunability, an important feature in accelerator-based THz sources. A compromise between maximal tunability and minimal slippage is achieved by increasing the beam energy to detune from group resonance. Figure 1 shows the phase resonant frequencies for the first three modes of a 4.05 mm diameter circular waveguide as a function of beam energy in a helical undulator with period $\lambda_u = 3.2$ cm and normalized undulator parameter $K = 2.17$. In general, the phase resonance can be expressed by

$$\omega = ck_u\beta_z\gamma_z^2\left(1\pm\sqrt{1 - \frac{k_u^2 + k_\perp^2}{k_u^2\gamma_z^2}}\right) \tag{2}$$

where $\gamma_z^2 = \gamma^2/(1 + K^2)$. For energies above the group resonance condition, two frequencies can lase simultaneously due to the quadratic dispersion of the waveguide while no lasing can occur at lower energies. The frequency branches are bound above by the free-space phase resonance asymptote and below by the fundamental cutoff frequency of the waveguide. Significant frequency tuning is only achieved on the high frequency branch and a main challenge of a compact THz FEL lies in achieving adequate seeding at these higher frequencies. In most setups with pulsed high current beams from photoinjector guns, the low frequency branch will also emit coherently, thus resulting in a complex multifrequency radiation pulse.

Note that Fig. 1 also suggests that high frequency generation can be achieved by targeting zero-slippage at higher order modes instead of reducing waveguide dimensions. Though higher order modes have a weaker coupling, the ability to operate closer to zero-slippage with relaxed tolerances for full charge transmission through the undulator makes them attractive for extending the operational frequencies. Further, while a helical undulator geometry offers advantages such as enhanced FEL coupling as well as the generation of circularly polarized radiation, a planar geometry has an important advantage near zero-slippage operation as the waveguide aspect ratio can be chosen to maximize beam clearance in the deflection plane[29,30]. A planar geometry would also allow for the use of grooved or Bragg waveguide structures to reduce losses in the system[31,32]. Finally, a curved parallel

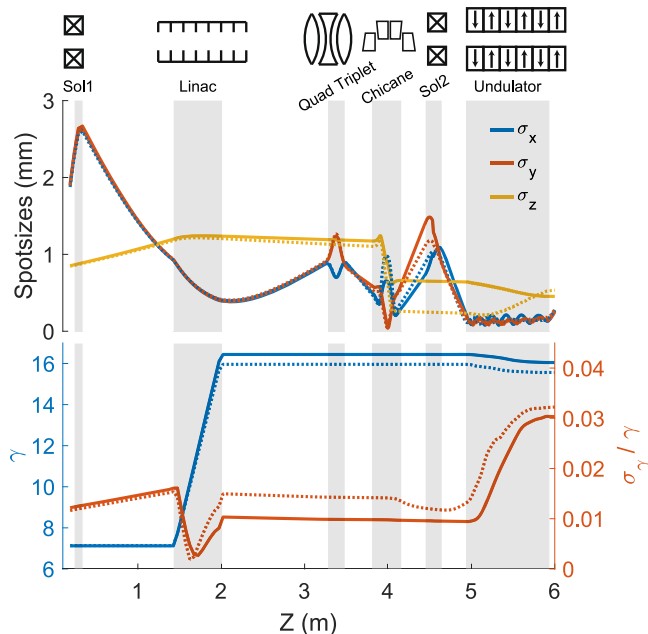

**Fig. 2 | Start-to-end particle tracking simulations along the UCLA Pegasus beamline.** Top) Evolution of the x,y,z rms moments of the particle distribution. Bottom) Energy and energy spread of the beam along the beamline. Solid and dotted lines show transport of beamlets and smooth longitudinal profile, respectively.

plate waveguide with adjustable gap can further improve tunability as it becomes possible to vary the zero-slippage frequency[33,34].

## Experimental setup

The experiment was conducted at the UCLA Pegasus photoinjector beamline[35]. A temporally shaped 260-nm laser pulse illuminates an alkali antimonide photocathode in an S-band radiofrequency (RF) gun generating electron bunches with charges in excess of 300 pC per pulse. A booster linac cavity is adjusted in phase and amplitude to both accelerate the beam up to 8.5 MeV total energy and imprint a tunable energy chirp for compressing the beam in the permanent magnet chicane. A quadrupole triplet focuses the beam into the chicane where the dipole entrance and exit angles are shaped to symmetrize the focusing in the horizontal and vertical planes. The second solenoid is used to match the strongly divergent beam into the undulator. Figure 2 shows start-to-end GPT simulations of the beam transport for the two different schemes of laser temporal shapes employed in the experiment, i.e. a sequence of well-separated beamlets and a smooth flat-top longitudinal profile represented by solid and dotted lines, respectively. While the linac phase was tuned to achieve maximal compression for the smooth distribution, in the beamlets case, we operated the linac at an under-compression phase to generate a multi-peak current density (see Fig. 7), which is the cause for the differences in bunch length evolution along the undulator.

**Beam generation.** One of the enabling components of this experiment is the capability to leverage the high quantum efficiency cathodes installed in the Pegasus photoinjector as these allow lossy pulse shaping techniques without compromising bunch charge. The Na-K-Sb cathodes for this experiment were grown at Cornell and transported to UCLA in a vacuum suitcase. They provide an order of magnitude improvement in QE compared to Cu photocathodes. A full characterization of the photoemitted beam characteristics is beyond the scope of this paper, but a maximum QE approaching 1 % in the UV and charges up to 500 pC were recorded at Pegasus.

In practice, the initial 100 fs FWHM laser pulse is shaped using different thickness birefringent $\alpha$-BBO crystals[36]. A 150-mm-long

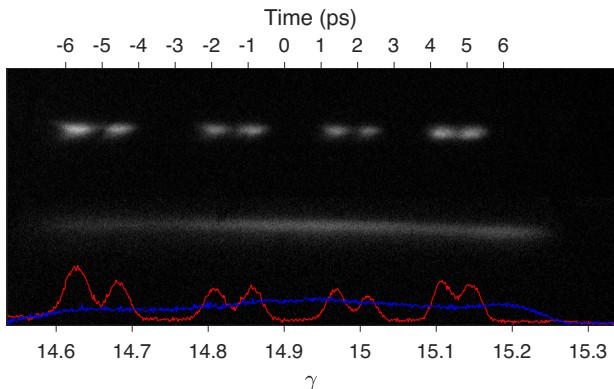

**Fig. 3 | Spectrometer measurements of beamlet and smooth charge distributions.** The laser is shaped with $\alpha$-BBO crystals and a fused silica rod. Energy is mapped to time using simulations calibrated to linac measurements.

dispersive fused silica rod can also be added when a smooth profile is desired. Stretching of the photocathode laser pulse reduces the beam current density in the gun significantly reducing space charge induced energy modulations. Most importantly, as described below, it is possible to generate a well separated train of beamlets to enhance the seeding at higher frequencies in the FEL. This is critical when the FEL is detuned from group resonance since a fully compressed single bunch no longer efficiently couples to the radiation. In this case, the slippage-dominated interaction instead favors a more advanced pulse shaping scheme with strong bunching from a multicycle current distribution.

Figure 3 demonstrates pulse shaping as measured on the energy spectrometer at low charge when the linac phase is set 40° off crest. The beamlet distribution is generated with a series of 8, 4, and 1 mm crystals (inducing 8, 4 and 1 ps delays at the cathode) where the 1 mm crystal is used to maximize the charge per beamlet without sacrificing beamlet separation. The smooth distribution utilized an additional 2 mm crystal with the 150-mm-long UV fused silica smoothing rod. For higher charges, the beam undergoes longitudinal space charge oscillations[26] and the energy distribution is not a good representation of the temporal profile.

**Permanent magnet chicane.** A compact, permanent magnet chicane was used to compress the beam within a short distance. The R56 design value of 6.4 cm allows for reaching full compression with a limited energy chirp, minimizing the reduction in FEL gain associated to the beam energy spread.

The chicane is comprised of four C-shaped permanent magnet based dipoles (shown in Fig. 4i inlay) with 8 mm gap and peak fields up to 250 mT[37,38]. The fields are modeled with the 3D magneto-static code RADIA[39] and fine tuned with magnetic shims. The large bending angle ($\theta_b = 37°$) limits the energy transmission to $15 < \gamma < 17$ for the nominal magnet offsets and results in significant transverse focusing. The pole angles ($\theta_1, \theta_2$) were chosen to equally distribute focusing between the transverse dimensions while maintaining zero dispersion in simulations. Note that the choice $\theta_2 > 90°$ was necessary to compensate for the simulated x-dependence in peak fields, seen in Fig. 4.

During the experiment, it was observed that the chicane did in fact exhibit a small negative energy dispersion, although at a level where beam matching into the undulator was not compromised. Figure 4ii shows data and simulations for the dispersion as measured on a screen 83 cm downstream of the chicane. Hall probe measurements performed after the chicane was removed from the beamline showed little x-dependence in the magnetic fields with the simulation discrepancy due to an incorrect value for the magnetic susceptibility of the carbon steel yoke. Corrected simulations show agreement with the measured dispersion data while suggesting minimal dispersion for $\theta_2 = 90°$.

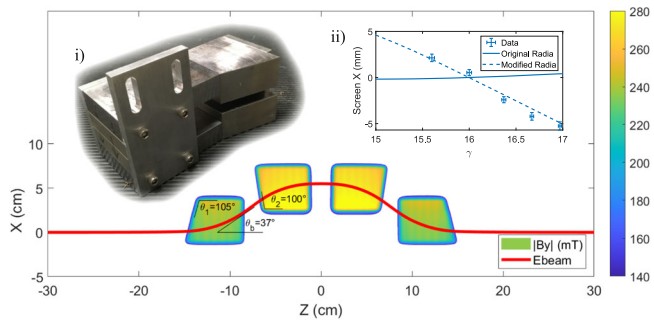

**Fig. 4 | Simulated chicane fields shown with beam trajectory.** i) A photo of an as built single dipole magnet. ii) Energy dispersion measurements (data points from 30 shots averages and corresponding error bars) from screen 83 cm downstream of chicane exit plotted against the original and modified RADIA chicane models.

Measurements of chicane focusing inferred from two downstream screens as a function of upstream steering align with the corrected simulations, demonstrating good agreement in the transverse dynamics.

**Undulator.** The helical undulator employed in the experiment is a section of the Theseus system[18,40], a double Halbach array of permanent magnets with 30 periods of length $\lambda_u = 3.2$ cm including an entrance and exit period. The magnetic field profile was tuned by adjusting the magnetic gap using longitudinal Hall probe scans targeting an initial peak field of 0.73 T to maximize the energy extraction from the relativistic beam. The end periods and field integrals were tuned on a pulsed-wire bench. By measuring at different transverse wire positions, corrections are also made for higher order quadrupole and sextupole moments in the off-axis field. This is particularly important for the beam energies used in the experiment as the radius of the helical trajectory in the undulator is 700–800 μm such that particle transmission through the system is largely affected by the behavior of the off-axis fields. The vacuum pipe (ID = 4.93 mm) and waveguide (OD = 4.76 mm) dimensions were chosen so the waveguide fit tightly inside the vacuum pipe which was tensioned on the pulsed-wire bench to optimize alignment between the waveguide and magnetic axis.

The FEL interaction was simulated using GPT-FEL[41], a custom element built on the GPT code framework[42] that self-consistently models the energy exchange between the particles and the electromagnetic field which is decomposed in a basis of frequency and spatial modes, naturally incorporating waveguide dispersion. Full start to end simulations are possible while seamlessly interfacing with cavity field maps, space charge routines, and other GPT functionality.

**Diagnostics.** After the undulator, radiation is collimated with an off-axis-parabolic (OAP) mirror and reflected out of the beamline while electrons pass through a 5 mm diameter hole in the mirror. A second OAP focuses the radiation onto a pyroelectric detector for pulse energy measurements. Figure 5 shows a measurement of THz pulse energy as a function of the nominal bunch charge for a smooth beam distribution. Since the charge measurement is before the undulator, it does not account for transmission losses in the waveguide. Below 50 pC, the energy grows quadratically (black dotted line) as expected for superradiant emission. As the charge is increased, space charge effects cause a decrease in bunching and an increase in transmission losses through the undulator, resulting in sub-quadratic energy growth. Note that these pulse energies have not been corrected for the estimated 35% loss in THz transport. At the maximum injected charge of 350 pC, we measured 18.5 uJ at the detector (estimated 28 uJ before losses) with 160 pC transmitted through the 1-meter undulator in rough agreement with the expected quadratic dependence. The efficiency in this experiment is notably lower than previous results at the zero-slippage condition[18] due to the reduced initial bunching factors at

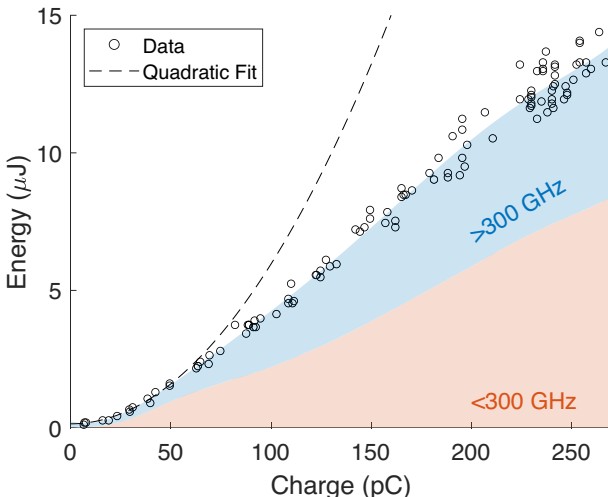

**Fig. 5 | Measured THz energy as a function of injected bunch charge.** Energy is recorded with a pyro detector and charge is measured with an integrating current transformer located between the gun and the linac. Shading shows simulated energy content of the high and low frequencies. The black dotted line is a fit to the data for charges below 50 pC where transmission losses are negligible, showing the expected quadratic growth.

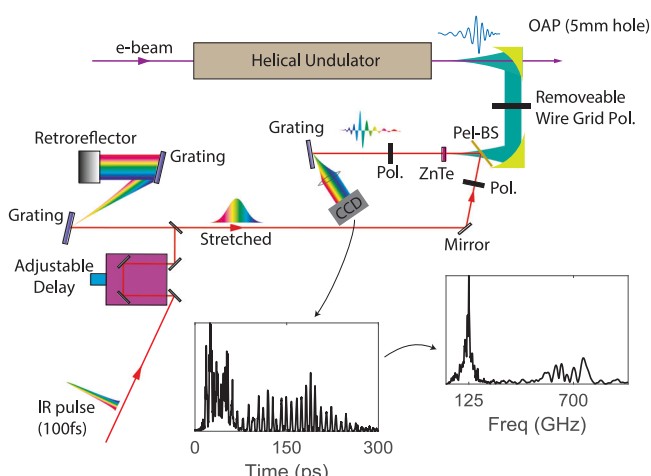

**Fig. 6 | Schematic of the EOS measurement.** Inlays show the measured temporal intensity and computed spectrum for the beamlet distribution at $\gamma = 16.3$.

higher frequencies, and, even more importantly, to the slippage effects which stretch the radiation pulse and lower the decelerating field amplitude in the undulator.

In Fig. 5 we also show simulated results for the energy content at frequencies above and below 300 GHz, which the pyro-detector can not distinguish. Integrating photons over the entire spectral range completely hides the complexity of the radiation pulse structure. To better understand the behavior of the system, the pyro-detector is replaced with a ZnTe crystal and a single-shot, cross-polarized electro-optic sampling (EOS) measurement of the temporal profile of the field[43] is implemented as shown in Fig. 6. In this measurement, the time-trace of the THz FEL electric field is encoded in the spectrum of a synchronized chirped IR pulse and can be retrieved in a single shot which is critical to eliminate issues related to timing jitter which severely limit the resolution above 500 GHz. In the time-traces of the pulse, corresponding to an input energy of 8.1 MeV, it is immediate to observe a leading high frequency pulse followed by a low frequency tail. The high frequencies are at the limit of measurement resolution

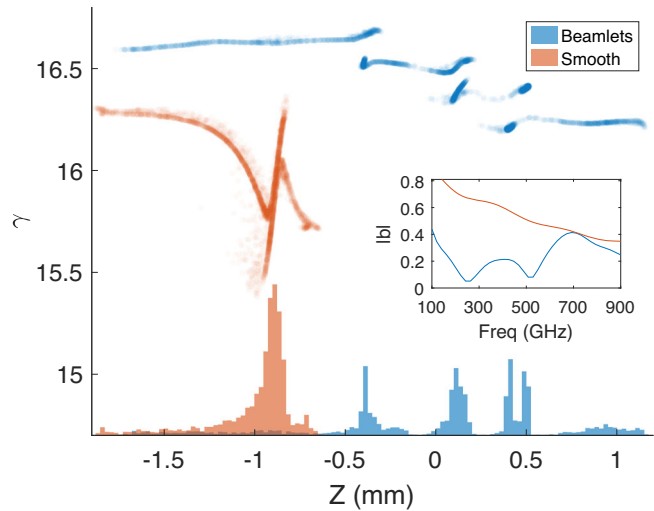

**Fig. 7 | Simulated longitudinal phase space for smooth and beamlet distributions.** The corresponding temporal current profiles are visualized with histogram projections and the inlay shows spectral bunching factors, a measure of beam coherence, at the undulator entrance.

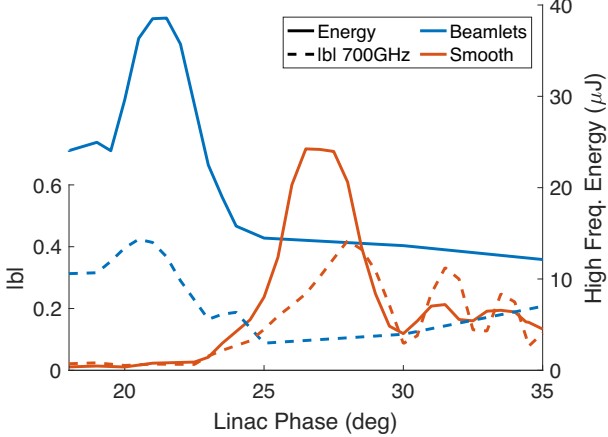

**Fig. 8 | GPT simulations of longitudinal compression.** The energy and magnitude of the bunching factor for smooth and beamlet distributions are plotted versus the RF linac phase where 0° corresponds to the maximum energy setpoint. The bunching is computed for the 700 GHz phase resonance at the undulator entrance.

while the low frequencies experience greater slippage and fill a majority of the waveform. Due to the large dispersion (and hence delay) between high and low frequency components in the waveguide, the entire radiation pulse covers a window of nearly 200 ps, so that a complete measurement with sufficient resolution requires stitching multiple measurements, each covering a 65 ps time window. The timing is adjusted with a calibrated delay stage and ~30 single-shot EOS images are averaged at each position after subtracting temporal jitter (Supplementary Fig. 2). A Fourier transform of the EOS trace shows peaks in the spectral response at 125 GHz and 700 GHz, matching the analytical predictions for the resonant frequencies at the injected beam energy from Fig. 1.

## Prebunching

In order to realize a compact FEL, it is important to seed the FEL interaction. In the absence of an external radiation pulse, the beam must be manipulated to generate a large initial bunching $b$, defined as the amplitude of the normalized Fourier transform of the longitudinal current profile at a given frequency. When injected in the undulator, this beam will coherently emit radiation, with a superradiant power scaling proportional to the square of the number of particles, providing a strong radiation field for the particles to interact with. At long wavelengths, this form of seeding can be achieved by compression to sub-wavelength scale, but the challenges in reaching sub-ps bunch lengths with >100 pC charges at moderately relativistic energies limit this scheme as frequencies approach 1 THz.

More importantly, detuning from group resonance introduces slippage that erodes the feedback between radiation and ultrashort beams. In order to compensate these shortcomings, we employed the prebunched, multipeak current density distribution generated from beamlets using α-BBO crystals. Figure 7 shows a comparison of the longitudinal phase spaces for the two distributions at the undulator entrance with an arbitrary positional shift added for visual clarity. The effects of non-linearities in energy chirp and strong longitudinal space charge after compression can be seen in the folded z-shape of the smoothed distribution. The beamlets are generated using 8, 4 and 1 mm α-BBO crystals (Supplementary Fig. 1) which have been found to maximize the bunching after the non-linear longitudinal space charge oscillations along the beamline[26]. Gun and linac phases are adjusted to tune the beamlet overlapping, resulting in a peaked current density with partially tunable period. It has been pointed out[44,45] and experimentally

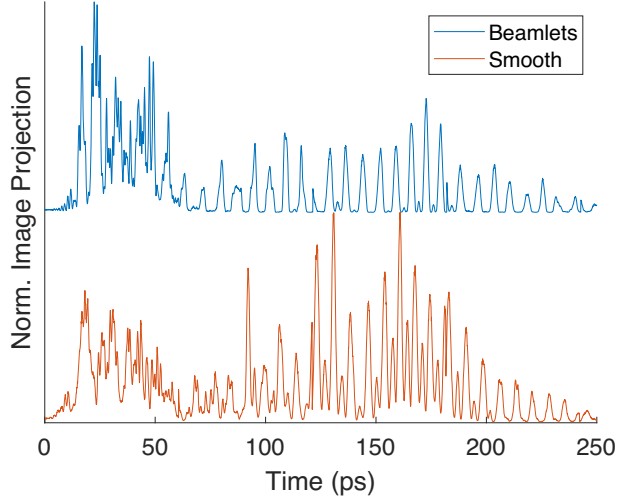

**Fig. 9 | Normalized EOS measurements of temporal intensity for smooth and beamlet distributions at $\gamma = 16.3$.**

demonstrated[46] that the emission of radiation can be significantly enhanced by using proper energy-phase correlation. In practice, the RF phases are fine-tuned as optimal bunching at the undulator entrance does not necessarily correspond to maximal emission due to the additional evolution of the longitudinal phase space along the undulator[45].

Figure 8 plots the simulated bunching and high frequency energy for each distribution at various compressing linac phases. Though both distributions have a maximal bunching factor above 40% at phase resonance, the beamlet distribution reduces the total energy spread and is more robust to slippage introduced off group resonance, leading to nearly double the high frequency content of the smooth distribution. The smooth distribution requires stronger compression and thus operates at a higher linac phase. When overcompressed, the nonlinear chirp results in two density peaks which causes oscillations in the bunching factor as their separation increases. Note that pulse energies are larger in ideal simulations than experiment due to optimal beam transport and alignment. Both the beamlets and smooth beam provide strong emission at the low frequency, generating 23 μJ and 31 μJ at 125 GHz, respectively.

Figure 9 shows normalized lineouts of EOS measurements proportional to field intensity where gun and linac phases were tuned to

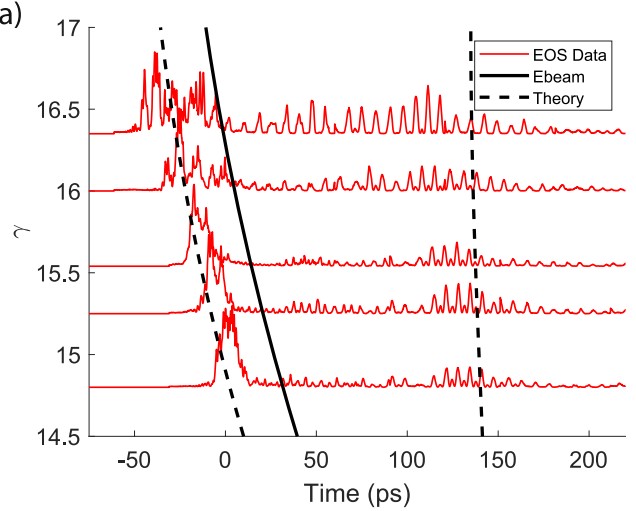
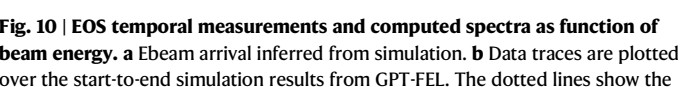
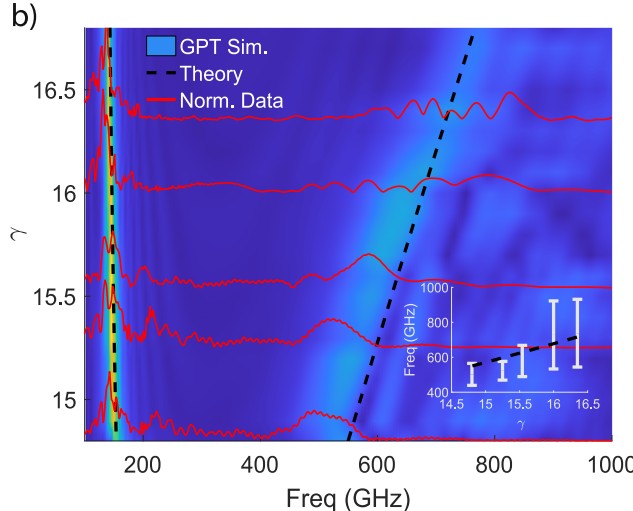

**Fig. 10 | EOS temporal measurements and computed spectra as function of beam energy. a** Ebeam arrival inferred from simulation. **b** Data traces are plotted over the start-to-end simulation results from GPT-FEL. The dotted lines show the phase resonances from the analytical theory. Inlay shows tunability of the central peak in the high frequency branch and the associated rms spectral bandwidth (white) over theory (black).

optimize the high frequency content. As expected, the beamlet distribution generates a waveform with a larger relative high frequency response. By integrating the lineouts over each frequency regime, it is estimated that the energy ratio for high frequency to low frequency is improved by a factor of 2.5 consistent with the simulation prediction.

## Frequency tuning

While the zero-slippage condition is rigorously satisfied at a single frequency for a fixed waveguide, different frequencies can be targeted when the beam energy is detuned above resonance. Temporal EOS measurements are shown in Fig. 10a as a function of beam energy where gun and linac phase are adjusted to optimize high frequency bunching for each measurement. The arrival times for the electron beam (solid) and phase-resonant frequencies (dashed) are simulated to account for the beam velocity/group velocities in the waveguide and for differences in beam propagation time to the undulator. Time zero is chosen as the high frequency arrival at $\gamma = 14.8$ for both measurement and simulation.

The measured waveform aligns well with the theoretical arrival of both resonant frequencies and the emission strength along the undulator can be indirectly inferred due to waveguide dispersion as radiation emitted near the entrance and exit of the undulator corresponds to the waveform intensity near the dashed and solid lines, respectively. The weak low frequency signal in the middle of the waveform can explained by a reduction in bunching factor and loss of charge transmission through the waveguide. At higher energies, charge is better transmitted through the system and both the low and high frequency generation is more uniform. While there is significant energy content in the low frequencies, the high frequencies experience less slippage and are more closely coupled with the longitudinal phase space dynamics.

Figure 10b shows the spectra calculated from the EOS measurements on top of GPT-FEL simulations and the analytic phase resonances for an effective waveguide diameter of 3.75 mm. As when operating the beamline, compression setpoints in simulation were chosen to optimize the bunching factor for the higher resonant frequency at the undulator entrance. The low frequencies line up well with simulation and theory with the bandwidth narrowing off resonance due to an increase in slippage. On the other hand, for the high frequency branch, there is a clear redshift at the lowest input beam energies both in the measurements and simulation which can be traced back to the large energy spread of the beam at the entrance of the undulator for

these cases. For $\gamma > 16$, the central frequency in the pulse is in better agreement with the analytical theory, but a clear increase in the bandwidth of the radiation and multiple spectral peaks can be observed. This can be in part due to the limitations of the diagnostics, which is affected both by absorption in the THz transport as well as the limitations in full spectral retrieval of the single-shot EOS technique[47]. At the same time, a wide, structured radiation spectrum is also consistent with the simulation, especially for the higher input energies and can be attributed to the difficulty of attaining and maintaining strong consistent bunching for the entire length of the undulator, leading to a broadening of the spectral form factor and a more complex structure of the emitted radiation. The inset more clearly shows tunability and bandwidth of the source as a function of beam energy.

## Outlook

In summary, we have demonstrated higher frequency generation and frequency tunability in a compact, prebunched waveguide THz-FEL by operating off zero-slippage resonance.

The main challenge of seeding higher frequencies in the presence of modest slippage introduced off resonance was addressed through laser shaping with $\alpha$-BBO crystals enabled by the high quantum efficiency Na-K-Sb cathode. In addition, the beamline was upgraded with a permanent magnet chicane and an additional solenoid to provide necessary compression and transverse focusing for beam transport. The spacing between the emitted electron beamlets can be tuned with the gun and linac cavities to sufficiently seed the compact FEL at different frequencies with a multicycle current distribution robust to FEL slippage.

Start to end simulations were conducted entirely in GPT with transverse and longitudinal space charge routines as well as custom field maps. Waveguide dispersion is naturally included in the modal frequency-spatial decomposition within the GPT-FEL module.

In addition to energy measurements with a pyroelectric detector, we employed a single-shot, cross-polarized electro-optic sampling measurement to resolve the temporal structure of the dual-frequency lasing of the FEL. Analysis of the spectra confirmed frequency tuning across the experimental energy range in agreement with theory. In principle, the complex structure of the radiation pulse could be tailored for 2D THz spectroscopy experiments[48]. This could be done either by splitting and recombining the pulses using wavelength sensitive optical elements or simply controlling the delay along the spectrum with dispersive optics[49].

Further paths of investigation include lasing at higher order waveguide modes to increase the frequency of the zero-slippage condition and adopting a planar geometry to reduce alignment tolerances and improve charge transmission.

## Methods

### Pegasus beamline

A frequency-tripled Ti:Sa laser system generates 260-nm UV pulses with up to 100 μJ of energy and 100 fs FWHM pulse length. Birefringent $\alpha$-BBO crystals, rotated 45° relative to the laser polarization directions, consecutively split the laser along fast and slow axes with a 1 ps delay per millimeter of crystal thickness at 260 nm. A 150 mm fused silica rod with 200 fs$^2$ mm$^{-1}$ group velocity dispersion at 260 nm stretches the laser pulse to 0.8 ps FWHM. The laser is focused onto a multi-alkali (Na-K-Sb) photocathode with a 3 mm diameter uniform spotsize and the 1.6-cell S-band RF gun accelerates the electron beam to an energy of 3.6 MeV. Bunch charge is measured non-destructively for each shot with a 5% accuracy with a turbo integrating current transformer (ICT) before entering an 11-cell S-band high-shunt-impedance linac with peak gradient of 20 MV m$^{-1}$. The gun solenoid and a quadrupole triplet focus the beam into the permanent magnet chicane. An additional solenoid, 4.55 m from the cathode, focuses the beam to its matched spot size of 150 μm at the undulator entrance, 4.94 m from the cathode. The gun and linac RF phases are tuned to induce energy chirp for ideal compression of the longitudinal distribution at the undulator.

### Permanent magnet chicane

The chicane measures $180 \times 60 \times 300$ mm$^3$ and is installed in a vacuum box on the beamline. Each magnet consists of two $50 \times 50 \times 12.5$ mm$^3$ NeFeB magnets in a C-shape yoke of 1006 High Carbon Steel with 8 mm full gap. The outer and inner chicane magnets are offset −13 mm and −50 mm in $\hat{x}$ relative to the beam axis to target an energy acceptance of $15 < \gamma < 17$. Resin 3D printed spacers hold magnets at their nominal relative positions. Pole angles ($\theta_1 = 105°$, $\theta_2 = 100°$) are chosen to minimize dispersion and equally distribute focusing in the transverse dimensions. Peak fields (roughly 250 mT) are tuned offline with iron shims and online with a motorized shunt across the inner chicane magnets. The chicane produces an R56 of 6.4 cm with maximum beam deflection of $\theta_b = 37°$. Chicane focusing was measured with raster scans of a steering magnet 26 cm upstream of the chicane entrance, imparting a 5 mrad A$^{-1}$ kick with beam position measured on two screens, 15 cm and 83 cm downstream of the chicane exit (Supplementary Fig. 4).

### Theseus undulator

The helical undulator is comprised of two permanent magnet, Halbach arrays offset a quarter period relative to each other. There are 30 periods of length $\lambda_u = 3.2$ cm including entrance and exit periods for an overall length of 96 cm. The nominal on-axis field profile is optimized with GPT-FEL beam simulations and tuning adjustments are guided by longitudinal hall probe scans. End periods and field integrals are tuned with pulsed-wire measurements. The 0.73 T peak field at the undulator entrance is obtained at a 7 mm magnet gap. The vacuum pipe has a 6.35 mm outer diameter and 4.93 mm inner diameter. The copper waveguide has a 4.76 mm outer diameter and a nominal 4.05 mm inner diameter, fitting snugly inside the vacuum pipe. In order to align the magnetic and waveguide axes, the vacuum pipe is tensioned with extra long vacuum screws (simultaneously securing the flange and tensioning against the undulator strongback) on the pulsed-wire table while the wire is centered in the waveguide. The undulator is then aligned to the green reference laser for the beamline. The undulator support allows fine adjustment of position and angle. Beam transmission is optimized with a pair of steering magnets before the undulator.

### Simulation methods

Chicane fields are simulated with the 3D magnetostatics code RADIA. Objects are subdivided into 5-mm blocks and chicane fields are evaluated on a 1-mm-spaced lattice for incorporation as a fieldmap into GPT beam simulations. Simulated beam trajectories in RADIA and GPT were confirmed to be identical.

The experiment requires an FEL simulation capable of modeling waveguide dispersion and 3D space charge. To this end, we utilize the GPT-FEL module which models the radiation field as a basis of frequency and spatial modes, with evolution determined by energy conservation with the beam. Start to end beam simulations with longitudinal and transverse space charge can then be performed entirely in GPT with chicane and undulator fieldmaps. Simulations were run for 5000 particles using 121 frequency modes equally spaced between 80 GHz and 1 THz for the fundamental TE11 mode. Higher order modes have no phase resonance and were confirmed to have negligible effect. Copper waveguide power attenuation is <1 dB m$^{-1}$ for frequencies below 800 GHz.

### Terahertz transport and diagnostics

The waveguide extends a few centimeters beyond the undulator to the focus (76.2 mm) of a gold-coated OAP with a 5-mm diameter hole for beam transmission. Waveguide outcoupling results in a theoretical 13% loss. Collection losses of roughly 20% occur at high and low frequencies on the OAP due to its 5-mm hole and 50-mm diameter. The collected radiation is collimated and reflected normally through a z-cut quartz THz window (<10% absorption) where a second OAP focuses the THz to an interaction point for energy or temporal measurements. The pulse energy is measured by inserting a 9 mm diameter pyroelectric Gentec terahertz detector with $7 \pm 0.7$ nJ mV$^{-1}$ factory calibration at 1-mm wavelength. The temporal intensity profile is measured by inserting a ZnTe crystal as part of a cross-polarized electro-optic sampling (EOS) measurement. A stretched IR pulse (with chirped frequency) probes the birefringence of the ZnTe crystal induced by the THz electric field. A grating and lens image the frequency responses on a CCD camera for single shot measurements of a 65 ps time window with 1.3 ps resolution. Multiple images are spliced to reconstruct the entire waveform. Note that the cross-polarization scheme only yields the rectified electric field waveform. The spectra are computed taking the Fourier transform after the signal is unrectified by switching the sign of consecutive peaks.

## Data availability

All data are available from the corresponding author upon request. The data supporting the findings of this study are available within the paper and its Supplementary Information.

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

## Acknowledgements

This work was supported by DOE grant No. DE-SC0009914 (P.M.). The undulator construction has been carried out under SBIR/STTR DE-SC0017102 (A.M. and P.M.) and DE-SC0018559 (A.M. and P.M.). The UCLA-Cornell photocathode transfer was funded by NSF under Grant No. PHY-1549132 (J.M. and P.M.).

## Author contributions

A.F. carried out the measurements and analyzed the data. M.L., A.O., and Y.Y. helped with UCLA Pegasus beamline operation, including alignment, vacuum and controls. P.M. proposed and supervised the experiment. C.P. and J.M. developed and transported the gun photocathodes. A.M. participated in the TESSA development and is the principal investigator on one of the supporting grants. R.A. and T.H. were responsible for the undulator construction. A.F. and P.M. prepared the manuscript, which was revised and edited by all co-authors.

## Competing interests

The authors declare no competing interests.
