## [Peer Review File · Nature Communications]

REVIEWER COMMENTS

Reviewer #1 (Remarks to the Author):

The presented article looks interesting, both for the argument and for the obtained results, and I believe it deserves publication with minor revisions.

Applications of THz radiation is becoming more and more important in the last few years and rediscovering the long wavelength side of the FEL panorama opens a new scenario for the research in this field and the significance to the field is evident.

The work describes the development and the operation of a THz-FEL, operating in a configuration where the radiation is confined inside a wave guide during the interaction, providing a number of features not available in a standard free space FEL. The authors are successful in exploiting some of these features, obtaining two frequency lasing and optimizing the efficiency of the FEL operation by manipulating the parameters of the electron beam.

However, some more bibliographic information would need to be added, when describing the story and the technique used to realize THz FELs in the last 35 years.

The first THz FELs were realized between the end of the '80s and the beginning of the 90's.

But in the references only the activity of the USCB FEL, operating CW with an electrostatic accelerator, was cited, while in the early '90s there were other sources operating in the THz region with RF accelerators [Phys. Rev. Lett. 70, 928–931 (1993)]. Moreover the theory of waveguide FEL operation, was formulated in the same period and it's worth citing it, being it one of the basis of the presented work, and a deeper analysis of the content of such works could be useful to improve the paper itself. The term "zero slippage condition", cited in the work, was created by the researchers working on this issue in a paper describing the precise kinematic and dynamic properties of a waveguide FEL [Optics Communications, 80, 417–424 (1991)].

At the zero slippage condition the two waveguide resonant frequencies merge in a single one, resulting in a broader gain curve, proportional to $1/N^{1/2}$, instead of $1/N$. So it is wrong to declare that operating at zero slippage precludes FEL tunability (lines 52-53 and 133-135). On the contrary, tunability of the system is higher when operating close to this condition, and efficiency is doubled (the Renieri limit is $1/N$ instead of $1/2N$). Moreover it must be noted that, when the electron bunch starts to emit in the undulator, it loses energy, so that the resonance condition is no more fulfilled (unless a tapered waveguide is used), but the system continues to emit because the gain curve is broad, and it's broader in a waveguide FEL.

Regarding the possibility to extract most of the possible energy from the beam in a single passage (so avoiding the use of a resonator, and obtaining a very high energy extraction percentage), it's worth to notice that this has been obtained almost 20 years before the cited reference [Phys Rev Lett. 93, 264801 (2004)] and the theory under such result, based upon the Energy-Phase correlation of the electron beam, was formulated in the 90's (a resume can be found in [Infrared Physics & Technology, 40 (1999), 161-174]).

For the same reason we find strange the choice of citing very recent articles about the possible use of adjustable waveguide gap, while a large number of different advanced waveguide configurations were already tested in the '90s: tapered waveguides [Optics Communications 100, 131-136 (1993)], grooved [Optics Communications 155, 187–196 (1998)] and planar [Nucl. Instr. and Meth. in Phys. Res. A 407, 448-453 (1998)], with Bragg [Optics Communications 127, 288-294 (1996)] ...

An important part of the paper is devoted to an to manipulate with different techniques the

longitudinal phase space of the electron beam, in order to obtain the ideal conditions at the undulator entrance. Simulations are shown regarding this issue, but I believe that some more information about the way this manipulation is carried out, and the physical mechanisms involved, would be useful for the reader.

It must also be noted that the evolution of the longitudinal phase space has to be controlled all over the whole transit inside the undulator, because it has been demonstrated that maximum compression at the entrance non necessarily corresponds to the more efficient solution [again Phys Rev Lett. 93, 264801 (2004)].

Regarding the graph in fig. 5, the declared quadratic behavior under 100 pC charge does not look to be satisfied: at least for charges greater than 50 the behavior seems to be linear, and the low charge zone as reported on the graph is too small to determine the exact behavior. An enlargement of the low charge zone, reporting experimental data with error bars would better clarify the situation.

It's very interesting the way of getting the spectral content of the radiation, performing a Fourier transform of the time profile of the radiation beam. My only concern is related to the fact that being the system driven by a photocathode, it would be difficult to ensure that every electron bunch, and consequently radiation bunch, carries the same frequency content. Some more details should be given regarding this issue, i.e. reporting about the repeatability of the results over different shots.

In the prebunching paragraph is declared that seeding is required to initiate emission in a compact FEL. This statement should be softened: in fact for long wavelength FEL, where the electron bunch length is comparable with the emission wavelength, the emission occurs in the so-called "coherent spontaneous emission" regime, with a quadratic dependence over the bunch current. So the coherent spontaneous emission itself is more than sufficient to initiate emission. Anyway the proposed prebunching technique for sure produces positive effects, contributing to increase the efficiency.

Finally an editorial suggestion: it would be easier for the reader to find the figures in the same page where they are cited.

Reviewer #2 (Remarks to the Author):

Summary:

The manuscript presents a method for generating coherent THz pulses from a single compressed or pre-modulated electron bunch. The result has significant potential and builds on the method of FEL lasing in a zero-slippage regime demonstrated previously by several of the coauthors.

General Comments:

The manuscript is well written and the results are presented clearly. One fundamental issue that has not been addressed is what kind of applications would benefit from the resulting 2-color THz emission. While the authors have described general applications in THz science, they have not offered any applications that would benefit specifically from the results presented here. One possibility could be to drive two different resonant modes of some material simultaneously. More intriguing may be in driving two coupled modes, which could have interesting effects, but I offer this only as a hypothetical. It is more often that the interest in 2-color experiments is related to pump-

probe type of measurements, but this would require a method of splitting the two spectral distributions in order to control the relative delay. Adding something in the context of benefits to specific types of applications would improve the impact of the results here beyond just the novelty of this method.

Some minor issues that should be addressed are noted in the detailed comments below.

Detailed Comments:

Fig. 2: The description in the text and the caption conflict regarding solid vs. dashed lines representing the smooth or beamlet profiles.

Fig. 2 (top): It appears that in the undulator, one type of profile compresses longitudinally (solid = smooth?), while the other stretches (dotted = beamlets?). Why is the longitudinal effect reversed for the different profiles?

Fig. 2 (bottom): The energy gain in the linac is greater for one profile (solid = smooth?) than the other profile (dotted = beamlets?). Meanwhile, the energy spread is greater for the higher energy profile (it appears that this relative energy spread, i.e. $\Delta E/E$, right?). Why do the energy and energy spread differ for the two profiles?

Fig. 3: Why have the authors chosen to separate the beamlets as shown in Fig. 3? Instead of spacing all of the beamlets evenly (e.g. using a 1mm, 2mm, and 4mm α -BBO), the beamlets are in pairs with a small spacing within each pair and a larger spacing between pairs. Does this “pair-type” spacing provide a higher bunch form factor than an evenly spaced set of beamlets for the high (or low) frequency branch of the FEL phase resonance? The authors should clarify why they have chosen this “pair-type” of beamlet spacing compared to evenly spaced beamlets.

Line 247: The notation $\gamma \in [15,17]$ may not look ideal. Depending on the final typesetting, the square brackets may be mistaken as a reference. An alternative may be $15 < \gamma < 17$. (Note: \in is “element of” symbol in the manuscript)

Fig. 6: There is a lot of substructure (i.e. multiple peaks) in the spectrum of the high frequency emission. Is there an explanation of the origin for this substructure? Is it consistent shot-to-shot?

Fig. 7: In the main plot, the charge distribution is plotted in energy vs. time, but the lower part of the main plot shows # of electrons (in bins) vs. time, correct? A secondary y-axis is not really necessary, but this distinction should be made clear in the caption.

Fig. 7 (inset): It is a bit surprising that the bunch form factor for the beamlet distribution is not larger than the smooth distribution at 700 GHz. It would also be useful to see how the bunch form factor compares at the low frequency branch (i.e. around 120 GHz). I suggest extending the range of the inset figure to include both branches.

Fig. 8: Where is this simulation measurement taken from (e.g. entrance of the undulator)? Also, how is the linac phase referenced? Is 0° defined as on crest? Please specify for clarity.

Line 404: Wording: It seems that the authors have changed the wording from beamlet and smooth to multicycle and single-cycle. If the meaning is the same, the wording should be kept consistent. Also, the wording of the statement “...provide large bunching at low frequency wavelength, generating 23 uJ and 31 uJ...” creates some problems. Since the authors give energy of the emitted

THz pulse, it is probably better to say "...provide strong emission...". Also, eliminate "wavelength".

Fig. 9: Please specify at the e- beam energy for the plotted measurement.

Line 432: The meaning of "...emission strength along the undulator can be indirectly inferred from the waveform intensity when moving from the outer dashed lines to inner solid line." is not completely clear. Is the idea that the bunch compression is improved at higher beam energy, or perhaps better e- beam transport at higher beam energy? The authors should explain this better in the text.

Line 532: Same comment as line 247.

Line 547: Capitalize Halbach.

Line 611: Please give more detail on how multiple EOS traces are stitched/spliced together. How is ensured that the time bases match from trace to trace. Also, while the full EOS measurement in the time-domain requires stitching multiple traces together, is each trace one single-shot measurement or are multiple single-shot traces averaged to provide better S/N in each trace?

Other: Can the authors provide a reference for the RADIA code?

Reviewer #3 (Remarks to the Author):

The THz radiations based on relativistic electron beams, especially the pre-bunched electron bunch trains, have become a very important part of the high brightness THz sources. These kinds of THz sources normally have high pulse energy, high coherence, high polarization, narrow band and tunable frequency. They are irreplaceable pump or probe lights for some scientific researches. This paper, adding a circular copper waveguide in the undulator, gives a method to increase the coupling between the relativistic electron beams and electromagnetic fields in a single pass free electron laser experiment. The paper gives very detail description of the experiment setup and the electron beam got from the photocathode rf gun and linac. The experimental data of THz radiation energy with different bunch charge (for a smooth distribution beam) are also given in the paper. A quadratical growth below 100 pC, as expected for superradiant emission, is got.

The paper has very solid experimental data, and maybe the first experimental result of the single pass THz-FEL with a copper waveguide in the undulator. It is valuable and important in the area. Some improvements need to be made before publication:

- 1) The title of the paper is "Towards to higher frequencies ...", while in the paper the experimental and simulated results are lower than 1THz. To do a new experiment with higher electron energy or different undulator is not an easy thing. Some simulation results for different electron energy, different pre-bunching, different waveguide and undulator, will be very helpful.
- 2) A more detail analytical description of the zero-slippage-resonance corresponding to the experiment setup, especially with the dispersion property of the waveguide, will be good for readers.
- 3) I still can not sure the radiation of the pre-bunched electron beam here is a coherent synchrotron radiation or a superradiant emission. Some simulation results needed here.

Report of Referee 1

The presented article looks interesting, both for the argument and for the obtained results, and I believe it deserves publication with minor revisions.

Applications of THz radiation are becoming more and more important in the last few years and rediscovering the long wavelength side of the FEL panorama opens a new scenario for the research in this field and the significance to the field is evident.

The work describes the development and the operation of a THz-FEL, operating in a configuration where the radiation is confined inside a wave guide during the interaction, providing a number of features not available in a standard free space FEL. The authors are successful in exploiting some of these features, obtaining two frequency lasing and optimizing the efficiency of the FEL operation by manipulating the parameters of the electron beam.

However, some more bibliographic information would need to be added, when describing the story and the technique used to realize THz FELs in the last 35 years.

The first THz FELs were realized between the end of the '80s and the beginning of the 90's.

But in the references only the activity of the USCB FEL, operating CW with an electrostatic accelerator, was cited, while in the early '90s there were other sources operating in the THz region with RF accelerators [Phys. Rev. Lett. 70, 928–931 (1993)]. Moreover the theory of waveguide FEL operation, was formulated in the same period and it's worth citing it, being it one of the basis of the presented work, and a deeper analysis of the content of such works could be useful to improve the paper itself. The term "zero slippage condition", cited in the work, was created by the researchers working on this issue in a paper describing the precise kinematic and dynamic properties of a waveguide FEL [Optics Communications, 80, 417–424 (1991)].

We thank the reviewer for pointing out this important prior art in the field of RF accelerator driven THz FELs. This was an oversight on our part and we fully recognize the strong relevance and impact of this literature in our current research directions. Fundamentally, our work builds on these early efforts in understanding the dispersion characteristics of a waveguide FEL and the role of pre-bunching in the radiation generation. The main difference is that we are able to leverage the last two decades of progress in photocathode-based electron sources to increase our ability to shape and prepare the beam in order to maximize the radiation content. Photoelectrons are also inherently synchronized (at the sub-ps level) with the drive laser allowing us to employ an electro-optic sampling based technique to directly measure the complex THz waveform in the time-domain.

We have modified the introduction to emphasize the strong relevance and impact of this prior-art literature in our current work.

“A waveguide can be used to transversely confine the field along the undulator, and the slippage can be eliminated by choosing waveguide dimensions such that the subluminal radiation group velocity is matched to the average longitudinal electron beam velocity \cite{doria1991kinematic}. As already recognized in the early ‘90s \cite{ciocci1993operation}, the introduction of the waveguide and the zero slippage condition enables the use of RF-compressed, high current density beams, short compared to the radiation wavelength, for seeding and the generation of large-bandwidth, few cycle THz pulses with intense fields.”

In response to another referee comment, we have also expanded our summary discussion of the “zero-slippage” condition which was mainly covered in the original work of the Optics Communications paper (now correctly cited in line 44), describing analytically the dispersion properties of the waveguide (including the dependence on the waveguide radius for the TE₁₁ mode) as well as the analytical expression for the quadratic resonance curves shown in Fig. 1.

At the zero slippage condition the two waveguide resonant frequencies merge in a single one, resulting in a broader gain curve, proportional to $1/N^{1/2}$, instead of $1/N$. So it is wrong to declare that operating at zero slippage precludes FEL tunability (lines 52-53 and 133-135). On the contrary, tunability of the system is higher when operating close to this condition, and efficiency is doubled (the Renieri limit is $1/N$ instead of $1/2N$).

The reviewer brings up an important point which indicates the need to make a better distinction regarding the range and degree of frequency tunability. It is true that compared to free-space, the broader gain curve at zero-slippage supports FEL operation in a wider frequency range by using different seeding, either pre-bunching or slightly detuned beam energies. However, the peak of the gain is centered around the radiation frequency and beam energy that satisfy group and phase velocity matching conditions, which are uniquely determined for a given waveguide and undulator. When the beam energy is increased further though, the FEL is no longer operated at zero-slippage, but instead resonates at two distinct frequencies of which the high frequency is strongly energy dependent. This operation mode (utilized in the experiment) allows the frequency to be tuned over nearly an octave at the cost of introducing limited FEL slippage.

An ideal tuning would allow zero-slippage operation over a wide range of frequencies by changing the waveguide parameters (i.e. adjustable gap). This is not employed in the experiment, but was mentioned as a future path of research along with resonance with higher waveguide modes.

We have edited the text to make this clear:

“In this regime, the gain curve is significantly broader than a typical free-space FEL, the interaction length is extended, and 10 % energy extraction efficiency has been previously demonstrated \cite{fisherNature2022}. The additional constraint on experimental parameters restricts zero-slippage resonance to one single frequency for a given undulator and waveguide aperture, undermining FEL tunability....To further extend the frequency range accessible by the FEL, it is possible to purposefully detune the beam energy from the zero-slippage condition, striking a compromise between maximal tunability and minimal slippage in the interaction.”

Moreover it must be noted that, when the electron bunch starts to emit in the undulator, it loses energy, so that the resonance condition is no more fulfilled (unless a tapered waveguide is used), but the system continues to emit because the gain curve is broad, and it's broader in a waveguide FEL.

The referee is absolutely correct. The broader gain curve naturally supports larger energy losses due to the radiation emission from the particles. Still, it is important to mention the possibility of tapering the waveguide (or the undulator) in order to maintain the resonant interaction as the beam loses energy as suggested in the 1993 Optics Communications.

Line 143: Both undulator field strength and waveguide parameters could in principle be tapered to maintain resonance with the decelerating electrons \cite{dipace1993taperedwaveguide} along the interaction.

Regarding the possibility to extract most of the possible energy from the beam in a single passage (so avoiding the use of a resonator, and obtaining a very high energy extraction percentage), it's worth to notice that this has been obtained almost 20 years before the cited reference [Phys Rev Lett. 93, 264801 (2004)] and the theory under such result, based upon the Energy-Phase correlation of the electron beam, was formulated in the 90's (a resume can be found in [Infrared Physics & Technology, 40 (1999), 161-174]).

We have added references to the theory and experiments on the effects of energy-phase correlation when discussing the optimization of the pre-bunching by fine-tuning the cavity settings, noting, as pointed out by the referee, that optimizing the bunching at the undulator entrance does not directly correspond to maximal emission.

Line 431: "It has been pointed out \cite{doriaPRL1998, gallerano1999} and experimentally demonstrated \cite{doria2004enhanced} that the emission of radiation can be significantly enhanced by using a proper energy-phase correlation. In practice, the RF phases are fine-tuned as optimal bunching at the undulator entrance does not necessarily correspond to maximal emission due to the additional evolution of the longitudinal phase space along the undulator."

For the same reason we find strange the choice of citing very recent articles about the possible use of adjustable waveguide gap, while a large number of different advanced waveguide configurations were already tested in the '90s: tapered waveguides [Optics Communications 100, 131-136 (1993)], grooved [Optics Communications 155, 187-196 (1998)] and planar [Nucl. Instr. and Meth. in Phys. Res. A 407, 448-453 (1998)], with Bragg [Optics Communications 127, 288-294 (1996)]

We have added text and references to better describe the work done with different waveguide geometries. An important benefit of an adjustable gap waveguide is to enable tuning of the zero-slippage resonant frequency and enable strong emission over a wide range of frequencies.

Line 185: "a planar geometry has an important advantage near zero-slippage operation as the waveguide aspect ratio can be chosen to maximize beam clearance in the deflection plane \cite{doria1998planardesign,israeliTES2024}. A planar geometry would also allow for the use of

grooved or Bragg waveguide structures to reduce losses in the system \cite{1998grooved, 1996curved}.

An important part of the paper is devoted to manipulating with different techniques the longitudinal phase space of the electron beam, in order to obtain the ideal conditions at the undulator entrance. Simulations are shown regarding this issue, but I believe that some more information about the way this manipulation is carried out, and the physical mechanisms involved, would be useful for the reader.

The referee is correct in that the original manuscript didn't sufficiently elaborate on the details of the longitudinal phase space preparation and in particular, on the somewhat odd choice of crystal lengths employed in the experiment to maximize the high frequency content in the radiation. It should be noted that this choice does depend on the particular beamline configuration as it is strongly influenced by the transverse envelope evolution (in our beamline determined by existing apertures, for example the 8 mm diameter linac irises) as well as the bunch charge and target frequency.

In order to provide more details about the pre-bunching scheme utilized in the experiment, we have added a dedicated section and a comparison figure in a Supplementary Information document. Specifically, we show the evolution of the longitudinal phase spaces for 8,4,1 mm and 8,4,2 mm crystals at different points along the beamline. Looking at the final beam distribution and the associated bunching factor in the relevant frequency range, the choice of using the 8,4,1 mm combination appears to have a slight advantage, while still very convenient from the experimental point of view and was therefore adopted in the experiment.

It must also be noted that the evolution of the longitudinal phase space has to be controlled all over the whole transit inside the undulator, because it has been demonstrated that maximum compression at the entrance non necessarily corresponds to the more efficient solution [again Phys Rev Lett. 93, 264801 (2004)].

This has been addressed by the previous answer regarding the maximization of single pass emission. For convenience, we repeat the added text.

Line 431: "It has been pointed out \cite{doriaPRL1998, gallerano1999} and experimentally demonstrated \cite{doria2004enhanced} that the emission of radiation can be significantly enhanced by using a proper energy-phase correlation. In practice, the RF phases are fine-tuned as optimal bunching at the undulator entrance does not necessarily correspond to maximal emission due to the additional evolution of the longitudinal phase space along the undulator."

Regarding the graph in fig. 5, the declared quadratic behavior under 100 pC charge does not look to be satisfied: at least for charges greater than 50 the behavior seems to be linear, and the low charge zone as reported on the graph is too small to determine the exact behavior. An enlargement of the low charge zone, reporting experimental data with error bars would better clarify the situation.

The reviewer is correct that the quadratic dependence begins to fade after 50 pC, and this has been updated in the text. To clarify the interpretation of the data, we have added a dotted line in the figure showing the fit of the quadratic dependence for data below 50pC (see below). We then emphasize that the apparent linear dependence at higher charges is well explained by observed charge losses in the undulator. Specifically, we note that the quadratic fit roughly agrees with our measurement of 18.5uJ for 160pC transmitted through the undulator. This is now explained by the following text.

Line 340: “Below 50~pC, the energy grows quadratically (black dotted line) as expected for superradiant emission. As the charge is increased, space charge effects cause a decrease in bunching and an increase in transmission losses through the undulator, resulting in sub-quadratic energy growth.”

Line 348: “At the maximum injected charge of 350~pC, we measured 18.5~uJ at the detector (estimated 28~uJ before losses) with 160 pC transmitted through the 1-meter undulator in rough agreement with the expected quadratic dependence.”

In regards to the error bars, we note that each point in these plots corresponds to a single measurement shot on the beamline, so that the only error bars that could be added are related to uncertainties and systematic errors in the charge and energy measurement calibrations, which are noted in the Methods section of the paper.

It's very interesting the way of getting the spectral content of the radiation, performing a Fourier transform of the time profile of the radiation beam. My only concern is related to the fact that being the system driven by a photocathode, it would be difficult to ensure that every electron bunch, and consequently radiation bunch, carries the same frequency content. Some more details should be given regarding this issue, i.e. reporting about the repeatability of the results over different shots.

This is a very important point and we have added a section in Supplementary information to provide more details about the electro-optic sampling measurements. Since the beam structure is generated at the cathode by suitably shaping the photocathode driver laser, the THz waveform is phase-locked to the optical laser pulse. The largest variations in shot-to-shot EOS signal are observed in the relative time-of-arrival at the ZnTe crystal translating into a horizontal jitter in the retrieved waveforms. Note that this timing jitter is the main reason that prompted us to develop and utilize a single-shot EOS diagnostic as opposed to one where the relative delay between the IR laser and the THz radiation is scanned over many shots via a delay stage.

In order to correct for the time-of-arrival jitter we perform a simple realignment of the waveforms, maximizing the overlap between the peaks in post-processing. Incidentally, this provides us with a direct measurement of the time-of-arrival jitter yielding 0.8 ps rms. In this process, the outliers (< 10 % of the total number of shots) in the time-of-arrival distribution, corresponding to shots where relative time-of-arrival is more than 3 ps away from the mean, are removed from the data. Not surprisingly, the time traces for these shots are also the ones that have the largest differences from the average waveform, due to the variation of the RF phases and ensuing compression dynamics experienced by the beam. The temporal structure of the de-jittered traces surviving this cut presents minimal variation shot-to-shot as shown in the supplementary figure. This is to be expected since the radiation signal in our FEL system does not grow from noise, but is seeded with a well defined, prebunched structure.

In the prebunching paragraph is declared that seeding is required to initiate emission in a compact FEL. This statement should be softened: in fact for long wavelength FEL, where the electron bunch length is comparable with the emission wavelength, the emission occurs in the so-called “coherent spontaneous emission” regime, with a quadratic dependence over the bunch current. So the coherent spontaneous emission itself is more than sufficient to initiate emission. Anyway the proposed prebunching technique for sure produces positive effects, contributing to increase the efficiency.

The referee has a good point and we should have been clearer on this. In essence, we do consider compression of the beam to sub-wavelength scale as a form of seeding in our compact THz FEL. This occurs both due to the fact that the FEL process is initiated by a non-zero initial condition on one of the collective variables of the system (the beam bunching factor) and due to the coherent emission of radiation in the first few undulator periods which quickly provides a strong radiation field for the particles to interact with. These processes are intimately connected by the FEL equations and can't be fully distinguished.

We have revised the first paragraph of the Prebunching section to clarify our interpretation of seeding and the role of the “coherent spontaneous emission” in our compact FEL system.

“In order to realize a compact FEL, it is important to seed the FEL interaction. In the absence of an external radiation pulse, the beam must be manipulated to generate a large initial bunching b_b , defined as the amplitude of the normalized Fourier transform of the longitudinal current profile at a given frequency. When injected in the undulator, this beam will coherently emit

radiation, with a superradiant power scaling proportional to the square of the number of particles, providing a strong radiation field for the particles to interact with."

Finally an editorial suggestion: it would be easier for the reader to find the figures in the same page where they are cited.

This is an excellent suggestion and we plan to review the final typesetting with the editors to ensure proper placement for the figures.

Report of Referee 2

Summary:

The manuscript presents a method for generating coherent THz pulses from a single compressed or pre-modulated electron bunch. The result has significant potential and builds on the method of FEL lasing in a zero-slippage regime demonstrated previously by several of the coauthors.

General Comments:

The manuscript is well written and the results are presented clearly. One fundamental issue that has not been addressed is what kind of applications would benefit from the resulting 2-color THz emission. While the authors have described general applications in THz science, they have not offered any applications that would benefit specifically from the results presented here. One possibility could be to drive two different resonant modes of some material simultaneously. More intriguing may be in driving two coupled modes, which could have interesting effects, but I offer this only as a hypothetical. It is more often that the interest in 2-color experiments is related to pump-probe type of measurements, but this would require a method of splitting the two spectral distributions in order to control the relative delay. Adding something in the context of benefits to specific types of applications would improve the impact of the results here beyond just the novelty of this method.

The referee has an interesting suggestion which admittedly we had not considered in detail. It should be noted that the bandwidth of the high frequency and low frequency components of the radiation is significantly different. Also, while the high frequency is easily tunable with beam energy, the low frequency is not and mainly depends on the waveguide radius. We note that dispersion controlled optics (a layered mirror or section of waveguide with properly chosen radius) could be used to control the delay across the radiation spectrum. A way to split the two frequency components would be in the spectral domain (for example, if circular polarization is not required, a properly spaced wire polarizer would do that). Alternatively, a transverse spatial filter could also be used by exploiting the very different spot size evolution in a THz transport line. The split frequencies could then be delayed arbitrarily with respect to each other and used

in pump and probe experiments targeting either resonant interactions with elementary excitations in condensed-phase molecular systems or non resonant field-driven processes. In principle, one could design the system to interrogate a sample using the intense THz pulses as discussed for example in 2D THz spectroscopy (see for example K. Reimann et al. J. Chem. Phys. 154, 120901, 2021)

We added a sentence in the conclusion about the possibility of using the two resonant frequency bands from the compact FEL for pump and probe in multidimensional THz spectroscopy studies.

“In principle, the complex structure of the radiation pulse could be tailored for 2D THz spectroscopy experiments \cite{Reimann_2DTHzspectroscopy}. This could be done either by splitting and recombining the pulses using wavelength sensitive optical elements or simply controlling the delay along the spectrum with dispersive optics \cite{strecker2020}”

Some minor issues that should be addressed are noted in the detailed comments below.

Detailed Comments:

Fig. 2: The description in the text and the caption conflict regarding solid vs. dashed lines representing the smooth or beamlet profiles.

The referee caught an important typo. The erroneous ordering in the text has been changed.

Fig. 2 (top): It appears that in the undulator, one type of profile compresses longitudinally (solid = smooth?), while the other stretches (dotted = beamlets?). Why is the longitudinal effect reversed for the different profiles?

The smooth distribution (dotted lines) is maximally compressed at the undulator entrance and the beam length grows during the interaction. The beamlet distribution (solid lines) is undercompressed at the undulator entrance to create the multicycle charge distribution. The negative chirp leads to a small amount of apparent compression in the undulator.

The mislabeling clearly added to the confusion, but we also edited the text and included a reference to the figure of longitudinal phase spaces to make the bunch length evolution clearer to the reader.

Line 214: “While the linac phase was tuned to achieve maximal compression for the smooth distribution, in the beamlets case, we operated the linac at an under-compression phase to generate a multi-peak current density (see Fig. 7), which is the cause for the differences in bunch length evolution along the undulator.”

Fig. 2 (bottom): The energy gain in the linac is greater for one profile (solid = smooth?) than the other profile (dotted = beamlets?). Meanwhile, the energy spread is greater for the higher energy profile (it appears that this relative energy spread, i.e. $\Delta E/E$, right?). Why do the energy and energy spread differ for the two profiles?

Again, the mislabeling led to confusion.

The smooth distribution (dotted lines) requires stronger compression and hence operation of the linac cavity farther off crest, leading to a stronger energy chirp (larger energy spread) and a lower mean energy when compared to the simulation for the beamlet distribution (solid lines).

Fig. 3: Why have the authors chosen to separate the beamlets as shown in Fig. 3? Instead of spacing all of the beamlets evenly (e.g. using a 1mm, 2mm, and 4mm α -BBO), the beamlets are in pairs with a small spacing within each pair and a larger spacing between pairs. Does this “pair-type” spacing provide a higher bunch form factor than an evenly spaced set of beamlets for the high (or low) frequency branch of the FEL phase resonance? The authors should clarify why they have chosen this “pair-type” of beamlet spacing compared to evenly spaced beamlets.

The referee is correct in that the original manuscript didn't sufficiently elaborate on the details of the longitudinal phase space preparation and in particular on the somewhat odd choice of crystal lengths employed in the experiment to maximize the high frequency content in the radiation. It should be noted that this choice does depend on the particular beamline configuration as it is strongly influenced by the transverse envelope evolution (in our beamline determined by existing apertures, for example the 8 mm diameter linac irises) as well as the bunch charge and target frequency.

In order to provide more details about the pre-bunching scheme utilized in the experiment, we have added a dedicated section and a comparison figure in a Supplementary Information document. Specifically, we show the evolution of the longitudinal phase spaces for 8,4,1 mm and 8,4,2 mm crystals at different points along the beamline. Looking at the final beam distribution and the associated bunching factor in the relevant frequency range, the choice of using the 8,4,1 mm combination appears to have a slight advantage, while still very convenient from the experimental point of view and was therefore adopted in the experiment.

Line 247: The notation $\gamma \in [15,17]$ may not look ideal. Depending on the final typesetting, the square brackets may be mistaken as a reference. An alternative may be $15 < \gamma < 17$. (Note: \in is “element of” symbol in the manuscript)

The notation has been adjusted according to the referee's suggestion.

Fig. 6: There is a lot of substructure (i.e. multiple peaks) in the spectrum of the high frequency emission. Is there an explanation of the origin for this substructure? Is it consistent shot-to-shot?

This is an excellent observation. The shot-to-shot consistency of the temporal waveform (and hence of the spectral content) of the radiation is now discussed in the Supplementary Material document. The multiple peak structure in the high frequency data can in part be due to the limitations in the reliability of the single-shot electro-optic sampling at high frequencies. For example, Roussel et al. *Light: Science & Applications* (2022) points out that due to the large stretching of the IR probe pulse in the balanced detection EOS scheme, there are zeros in the transfer function so that information at certain frequencies is lost and the retrieved spectrum can have significant spurious structure.

At the same time, there is also significant structure present in the simulations, especially at the higher input beam energies. This is due to evolution of the beam bunching along the undulator, where the relatively large bandwidth of the waveguide interaction supports amplification at the near-resonant frequencies, complicating the THz radiation structure. We added a couple of sentences to comment on this aspect of the experimental data.

Line 512: “For $\gamma > 16$... a clear increase in the bandwidth of the radiation and multiple spectral peaks can be observed. This can be in part due to the limitations of the diagnostics, which is affected both by absorption in the THz transport as well as the limitations in full spectral retrieval of the single-shot EOS technique \cite{Roussel_PDEOS}. At the same time, a wide, structured radiation spectrum is also consistent with the simulation, especially for the higher input energies and can be attributed to the difficulty of attaining and maintaining strong consistent bunching for the entire length of the undulator, leading to a broadening of the spectral form factor and a more complex structure of the emitted radiation.”

Fig. 7: In the main plot, the charge distribution is plotted in energy vs. time, but the lower part of the main plot shows # of electrons (in bins) vs. time, correct? A secondary y-axis is not really necessary, but this distinction should be made clear in the caption.

The referee is correct, the lower part of the plot shows a histogram projection of the electron beam. The caption has been edited to read as

“Simulated longitudinal phase space for smooth and beamlet distributions and histogram projections to visualize the corresponding temporal current profiles.”

Fig. 7 (inset): It is a bit surprising that the bunch form factor for the beamlet distribution is not larger than the smooth distribution at 700 GHz. It would also be useful to see how the bunch form factor compares at the low frequency branch (i.e. around 120 GHz). I suggest extending the range of the inset figure to include both branches.

Full compression of the smooth distribution can indeed achieve the same bunching factor as the beamlet distribution at the undulator entrance. However, the beamlet distribution is much more effective at generating radiation with high frequency content as it is more robust to FEL slippage and has a smaller energy spread.

Following the referee suggestions we have extended the figure inlay to include the low frequencies (see below), The benefits of the beamlet distribution beyond the magnitude of the bunching factor are explained in the following paragraph:

Line 441: “Though both distributions have a maximal bunching factor above 40% at phase resonance, the beamlet distribution reduces the total energy spread and is more robust to slippage introduced off group resonance, leading to nearly double the high frequency content of the smooth distribution.”

Fig. 8: Where is this simulation measurement taken from (e.g. entrance of the undulator)? Also, how is the linac phase referenced? Is 0° defined as on crest? Please specify for clarity.

The figure caption has been updated to specify that bunching is computed at the undulator entrance and that a linac phase of 0 deg. corresponds to the maximum energy setpoint.

Line 404: Wording: It seems that the authors have changed the wording from beamlet and smooth to multicycle and single-cycle. If the meaning is the same, the wording should be kept consistent. Also, the wording of the statement "...provide large bunching at low frequency wavelength, generating 23 uJ and 31 uJ..." creates some problems. Since the authors give energy of the emitted THz pulse, it is probably better to say "...provide strong emission...". Also, eliminate "wavelength".

The original intent for the terms "single-cycle" and "multicycle" was to emphasize the impact of FEL slippage on the smooth and beamlet distributions. However, there were multiple locations (including Line 404) where we have been inconsistent in this regard and should have used the terms "smooth" and "beamlets", as quoted in the beam dynamics figure legends. These have been corrected and for the few direct references to seeding, the phrase "multicycle beam" has been replaced with the more appropriate "multi-peak current density".

To the second point, we have eliminated the typo "wavelength" and modified the text as suggested.

Fig. 9: Please specify at the e- beam energy for the plotted measurement.

A reference to the beam energy ($\gamma=16.3$) has been added to the figure caption.

Line 432: The meaning of "...emission strength along the undulator can be indirectly inferred from the waveform intensity when moving from the outer dashed lines to inner solid line." is not completely clear. Is the idea that the bunch compression is improved at higher beam energy, or perhaps better e- beam transport at higher beam energy? The authors should explain this better in the text.

The paragraph has been reworded to make it clearer that the emission strength along the interaction can be inferred due to the effects of the waveguide dispersion. Radiation emitted near the undulator entrance in fact will experience maximal dispersion delay and arrive near the dotted black lines computed from the group velocity of the two phase-resonant frequencies. Radiation emitted near the undulator exit will propagate for a shorter distance in the waveguide and therefore experience little dispersion arriving nearly simultaneously with the beam, denoted by the solid black line.

Line 481: "The measured waveform aligns well with the theoretical arrival of both resonant frequencies and the emission strength along the undulator can be indirectly inferred due to waveguide dispersion as radiation emitted near the entrance and exit of the undulator corresponds to the waveform intensity near the dashed and solid lines, respectively. "

Line 532: Same comment as line 247.

Done

Line 547: Capitalize Halbach.

Done

Line 611: Please give more detail on how multiple EOS traces are stitched/spliced together. How is ensured that the time bases match from trace to trace. Also, while the full EOS measurement in the time-domain requires stitching multiple traces together, is each trace one single-shot measurement or are multiple single-shot traces averaged to provide better S/N in each trace?

We have added the following sentence to clarify the measurement:

Line 384: "The timing is adjusted with a calibrated delay stage and ~30 single-shot EOS images are averaged at each position after subtracting temporal jitter (Supplementary)."

We have also added a section in Supplementary information where we provide more details about the temporal stability and reproducibility of the measurements.

Other: Can the authors provide a reference for the RADIA code?

The reference has been added (line 273).

Report of Referee 3

The THz radiations based on relativistic electron beams, especially the pre-bunched electron bunch trains, have become a very important part of the high brightness THz sources. These kinds of THz sources normally have high pulse energy, high coherence, high polarization, narrow band and tunable frequency. They are irreplaceable pump or probe lights for some scientific researches.

This paper, adding a circular copper waveguide in the undulator, gives a method to increase the coupling between the relativistic electron beams and electromagnetic fields in a single pass free electron laser experiment. The paper gives very detail description of the experiment setup and the electron beam got from the photocathode rf gun and linac. The experimental data of THz radiation energy with different bunch charge (for a smooth distribution beam) are also given in the paper. A quadratic growth below 100 pC, as expected for superradiant emission, is got.

The paper has very solid experimental data, and maybe the first experimental result of the single pass THz-FEL with a copper waveguide in the undulator. It is valuable and important in the area.

Some improvements need to be made before publication:

1) The title of the paper is “Towards to higher frequencies ...”, while in the paper the experimental and simulated results are lower than 1THz. To do a new experiment with higher electron energy or different undulator is not an easy thing. Some simulation results for different electron energy, different pre-bunching, different waveguide and undulator, will be very helpful.

The title of the paper refers to the improvement in the spectral reach in a compact THz waveguide FEL, in particular when compared to a pure ‘zero-slippage’ operation. In the latter case, the group and phase resonances are determined by the undulator and waveguide size and in order to access very high frequencies, the waveguide dimensions have to be reduced accordingly. In our scheme, by tuning the beam slightly off the zero-slippage, we are able to increase the emission frequency as well as the tunability of the system. We agree with the referee that the main challenge of operating such a compact, single-pass THz source at frequencies above 1 THz would be in achieving sufficient bunching

In order to answer this point and generalize the results discussed in the paper, we now present in the supplementary material section a simulation where, simply by assuming a higher accelerating gradient in our linac cavity, strong radiation emission at 1.5 THz is obtained. The compression dynamics are adjusted to generate a multi-peak current structure with the correct beamlet spacing at the entrance of the undulator.

Here is the supplementary material section on this.

“In order to demonstrate the possibility to extend the scheme discussed in the paper to higher frequencies, we present here a simulation study for a scaled version of our experiment where the zero-slippage resonance is increased to 0.4~THz and we target resonance at 1.5~THz with a beam energy of $\gamma = 23.2$ by artificially increasing the linac accelerating gradient to 32~MV/m. This field level is actually not feasible for the cavity currently installed on the beamline, but it is potentially achievable with a different S-band RF structure. The undulator parameters are unchanged and the waveguide radius is reduced to 1.42~mm.

A bunching factor of 0.33 can be achieved using the same set of 8,4,1~mm α -BBO crystals with similar beam transport and optimized gun phase (26°) and linac phase (17°). The injected longitudinal phase space is shown in Fig. \ref{fig:1p5THzcase} with the corresponding current profile and associated bunching factor. Given a nearly ideal transmission of 250~pC through the undulator, self-consistent GPT-FEL simulations indicate that a pulse energy of $28\text{~}\mu\text{J}$ is produced at and above 1.5~THz (spectrum shown in the inset of Fig. \ref{fig:1p5THzcase}b). The output longitudinal phase space shows clear evidence of the complex longitudinal dynamics resulting from the interaction of the THz radiation emitted by the beam with the electrons in the undulator. Of course, the smaller radius increases constraints on undulator tuning and waveguide alignment such that significant charge transmission might be difficult to achieve in practice.”

2) *A more detailed analytical description of the zero-slippage-resonance corresponding to the experiment setup, especially with the dispersion property of the waveguide, will be good for readers.*

We added a few citations on the original work describing the zero-slippage mode of operation as requested from Referee #1. In addition, given the importance to explain the experimental results in this paper, we also added a few sentences in order to analytically describe the dispersion properties of the waveguide (including the dependence on the waveguide radius for the TE₁₁ mode) as well as the analytical expression for the quadratic resonance curves shown in Figure 1.

Line 127: “The waveguide dispersion, $\omega^2/c^2 = k_z^2 + k_{\perp}^2$, alters the phase and group velocity of the radiation according to $v_p = \omega/k_z$ and $v_g = c^2 k_z / \omega$ where $k_{\perp} = 1.8412/R$ for the TE₁₁ circular waveguide mode with radius R .”

Line 159: “In general, the phase resonance can be expressed by $\omega = c k_u \beta_z \gamma_z^2 \left(1 \pm \sqrt{1 - \frac{k_u^2 + k_{\perp}^2}{k_u^2 \gamma_z^2}} \right)$ where $\gamma_z^2 = \gamma^2 / (1 + K^2)$.”

3) I still am not sure if the radiation of the pre-bunched electron beam here is a coherent synchrotron radiation or a superradiant emission. Some simulation results needed here.

The term superradiance in the FEL literature has been used multiple times with different meanings (weak superradiance, strong superradiance, superradiance ‘a la Dicke’). In all cases though, it refers to radiation emission proportional to the square of the beam charge so that our

experiment falls within the superradiant category. If the referee here wants to distinguish between simple coherent emission in the undulator and the FEL mechanism where the radiation acts back on the particles to preserve and increase the microbunching, our simulations show clearly visible ponderomotive potential buckets in the output longitudinal phase spaces, so in this sense this regime can be considered superradiant FEL amplification.

In essence, compression of the beam to sub-wavelength scale is used to seed our compact THz FEL. This occurs both due to the fact that the FEL process is initiated by a non-zero initial condition on one of the collective variables of the system (the beam bunching factor) and due to the coherent emission of radiation in the first few undulator periods which quickly provides a strong radiation field for the particles to interact with. These processes are intimately connected by the FEL equations and can't be fully distinguished.

Also in response to Referee #1 comments, the first paragraph of the Prebunching section has been edited to clarify our interpretation of seeding and the role of the "coherent spontaneous emission" in our compact FEL system.

"In order to realize a compact FEL, it is important to seed the FEL interaction. In the absence of an external radiation pulse, the beam must be manipulated to generate a large initial bunching b , defined as the amplitude of the normalized Fourier transform of the longitudinal current profile at a given frequency. When injected in the undulator, this beam will coherently emit radiation, with a superradiant power scaling proportional to the square of the number of particles, providing a strong radiation field for the particles to interact with."

REVIEWERS' COMMENTS

Reviewer #1 (Remarks to the Author):

As I previously stated in my first report, the article is interesting, both for the argument and for the obtained results. The interest of the scientific community for high power THz sources has grown in the last few years and THz FELs can play an important role in this scenario. In this framework the authors have fully exploited the features of the so-called THz waveguide FELs, providing solid results. All of the issue raised by me and by the other referees were considered and all of the required revisions were correctly implemented. Moreover additional explanations were added in a "Supplementary information" document, that better explains some of the more complex parts of the work.

The article is now clearer, and in my opinion it deserves publication, without any further modification.

Reviewer #2 (Remarks to the Author):

The authors have addressed all of my comments and questions. The addition of the supplementary information was particularly helpful. The manuscript is now suitable for publication in Nature Communications.

Reviewer #3 (Remarks to the Author):

I have no further questions and comments. The revised paper is good enough for publication.